# Microencapsulation of Carotenoid-Rich Extract from Guaraná Peels and Study of Microparticle Functionality through Incorporation into an Oatmeal Paste

**DOI:** 10.3390/foods12061170

**Published:** 2023-03-10

**Authors:** Lorena Silva Pinho, Bhavesh K. Patel, Osvaldo H. Campanella, Christianne Elisabete da Costa Rodrigues, Carmen Sílvia Favaro-Trindade

**Affiliations:** 1Departamento de Engenharia de Alimentos (ZEA), Faculdade de Zootecnia e Engenharia de Alimentos (FZEA), Universidade de São Paulo (USP), Pirassununga 13635-900, São Paulo, Brazil; spinholorena@gmail.com (L.S.P.);; 2Department of Food Science and Technology, College of Food, Agricultural, and Environmental Sciences, Ohio State University (OSU), Columbus, OH 43210, USA

**Keywords:** mechanical stress, thermal stress, β-carotene, lutein, stability, *Paullinia cupana*

## Abstract

The peels of guaraná (*Paullinia cupana)* fruit contain abundant carotenoid content, which has demonstrated health benefits. However, these compounds are unstable in certain conditions, and their application into food products can be changed considering the processing parameters. This study aimed to encapsulate the carotenoid-rich extract from guaraná peels by spray drying (SD), characterize the microparticles, investigate their influence on the pasting properties of oatmeal paste, and evaluate the effects of temperature and shear on carotenoid stability during the preparation of this product. A rheometer with a pasting cell was used to simulate the extrusion conditions. Temperatures of 70, 80, and 90 °C and shear rates of 50 and 100 1/s were the parameters evaluated. Microparticles with a total carotenoid content between 40 and 96 µg/g were obtained. Over the storage period, carotenoid stability, particle size, color, moisture, and water activity varied according to the core:carrier material proportion used. Afterward, the formulation SD1:2 was selected to be incorporated in oatmeal, and the paste viscosity was influenced by the addition of this powder. β-carotene retention was higher than that of lutein following the treatment. The less severe treatment involving a temperature of 70 °C and a shear rate of 50 1/s exhibited better retention of total carotenoids, regardless of whether the carotenoid-rich extract was encapsulated or non-encapsulated. In the other treatments, the thermomechanical stress significantly influenced the stability of the total carotenoid. These results suggest that the addition of encapsulated carotenoids to foods prepared at higher temperatures has the potential for the development of functional and stable products.

## 1. Introduction

Carotenoids are among the major classes of pigments found in tree leaves, fruits, and vegetables. Consumption of these compounds has been considered to bring health benefits. According to the literature, carotenoids may act as reducing agents for cancer [1,2,3], cardiovascular diseases [4,5] and macular degeneration [6], as well as antioxidants [7] and provitamin A [8]. The nutritive and coloring properties of carotenoids make them be an ideal food additive to develop functional products with desirable appearance.

The interest in recovering carotenoids from agro-industrial waste has increased, considering their high potential to enhance the valorization of by-products. Several studies [9,10,11,12] have reported the extraction of carotenoids from different vegetable wastes. However, guaraná residues, with potential uses in the production of soft drinks and food ingredient industries, have been rarely explored as a possible source of carotenoids in foods.

Carotenoids are highly susceptible to harsh external conditions, such as high temperature and shear, presence of oxygen, light, acidity, and pro-oxidant agents [13,14,15]. Encapsulation has emerged as a key approach for making the incorporation of these compounds in processed foods feasible. In the food industry, spray drying is the most widely used technology for the entrapment of bioactive molecules in the form of microparticles. The technique consists of atomizing a dispersion or emulsion, containing the component of interest and drying adjuvants, followed by its dehydrating forming microparticles. This encapsulation facilitates transport, prolongs shelf-life, and reduces the risk of carotenoid degradation during food processing [16,17].

Several additives are used as carrier materials to facilitate the drying, handling and application of these bioactive compounds. In addition, the property of these carrier materials can guarantee the stability of carotenoids against oxidation, storage conditions and processing in the food industry. Tuyen et al. [18] reported the effect of different concentrations of maltodextrin as carrier material and different inlet temperatures on color preservation, total carotenoid content and antioxidant activity of spray-dried gac powder. In addition, Hojjati et al. [19] investigated the influence of different concentrations of soluble soy polysaccharides on the properties of microcapsules loaded with canthaxanthin obtained by spray drying. Indeed, the encapsulation of canthaxanthin in this study resulted in a more significant storage stability of the samples. Highlighting the variety of carrier materials, Etzbach et al. [20] evaluated the effect of using different carriers such as maltodextrin, modified starch, inulin, alginate, gum arabic and cellobiose for spray drying golden blackberry juice rich in carotenoids. In this study, cellobiose’s proposed alternative carrier showed a high capacity to protect carotenoids from degradation processes by exposure to light, high temperature and oxygen, possibly due to a more compact particle wall and larger particle sizes.

The development of starchy products includes the investigation of new formulations and new processing techniques. However, typical technologies used to favor gelatinization and texturization of starchy products apply a combination of heat and shear over these products, which can be harmful to carotenoids. During gelatinization, starch swells, destabilizing its crystalline structure and loses birefringence. The main starch fractions, amylose and amylopectin, disperse and lead to the production of a paste [21,22]. After disruption of their granular structure, during processing, starch is subjected to mechanical stresses, mainly shear provoked by processing.

The rheological properties of starch solutions can vary according to factors such as amylose and amylopectin content, the presence of functional groups, and granularity. In the case of the current research, these factors can be influenced by adding microparticles loaded with carotenoids into the oat paste, which may change the paste’s rheological behavior. On the other hand, the processing of oat paste, which involves heat and shear conditions, can lead to the degradation of thermosensitive compounds such as carotenoids. From that, the proposal focused on understanding the effects of these conditions on microparticles rich in carotenoids, also evaluating the effectiveness of microencapsulation by spray drying. Oats have wide applications, and the carotenoids from the guarana by-product are still atypical but with great potential for exploration and use by the food industry.

To simulate a pasting environment where temperature and shear are applied, a rheometer equipped with a pasting cell was used to produce an oatmeal paste enriched with encapsulated carotenoids. Thus, this study was aimed to prepare a carotenoid-rich extract from guaraná peels, encapsulate this extract, investigate the influence of microparticles on the pasting properties of oatmeal, and evaluate the effect of thermo- and mechanical stress on carotenoid stability under different conditions.

## 2. Material and Methods

### 2.1. Materials

The Executive Commission for Cocoa Cultivation Planning, CEPLAC (Taperoá, Bahia, Brazil), provided the guaraná fruits. The fruit peels were removed from seeds and pulp followed by being washed with water. The reddish peels were dried in a convection oven (Marconi, MA035/1152) at 50 °C for 18 h [23]. They were then milled and stored under dark at −20 °C until analysis.

β-carotene (CAS 7235-40-7) and lutein (CAS 127-40-2) standards were purchased from Sigma-Aldrich (Saint Louis, USA). Analytical-grade ethanol (CAS 64-17-5), hexane (CAS 110-54-3), and acetone (CAS 67-64-1) were obtained from Fisher Scientific (Waltham, MA, USA).

The sunflower oil used during the guaraná peel extract preparation was from Cargill (brand Liza). The microparticles rich in carotenoids produced by spray drying (SD) were prepared using gum arabic as carrier material, obtained from Nexira, Brazil. Oat flakes (the Oat Quaker Company, Chicago, IL, USA) were purchased from a local supermarket at Columbus, OH, USA.

### 2.2. Carotenoid-Rich Extract Preparation

Carotenoid-rich extract preparation from guaraná peels was performed following [24], using ethanol as solvent, at a ratio of 1:10 (peel: solvent, *w*/*v*), during 4 h at 50 °C. The concentration of 3% of sunflower oil was added to the extract to reduce the carotenoid degradation detected during preliminary experiments. The final concentration was determined considering the liquid–liquid equilibrium for the mixture composed of oil and ethanol [25]. As follows, the material was concentrated using a rotary evaporator (TE-211 Tecnal, Piracicaba, Brazil) at 48 ± 2 °C to 20% of the initial volume. The concentrated extract was named guaraná peel extract (GPE).

### 2.3. Microencapsulation by Spray Drying

Formulations were prepared with a ratio of 1:2, 1:3, and 1:4 of concentrated ethanolic extract:gum arabic in aqueous solution (20% *w/v*), *v*/*v*, using Ultra-Turrax ^®^ IKA T25 (Labotechnic, Staufen, Germany) at 11,200× *g* for 3 min. The mixture was atomized according to Rocha et al. [26], with modifications. The spray dryer (Model MSD 1.0, Labmaq do Brasil, Ribeirão Preto, Brazil) was used coupled with a spray nozzle of 1.2 mm, with an inlet air temperature of 100 °C, an air drying speed of 2.5 m/s, a feed flow of 10 mL/min, and air pressure of 8.4 kgf/cm^2^.

### 2.4. Total Carotenoid Content

The microparticles were blended with hexane for 1 min and ultrasonicated for 20 min using an ultrasound bath, Branson 1800 (Branson Ultrasonics Corporation, Danbury, CT, USA) to extract carotenoids. Absorbances of extracts were measured at 450 nm wavelength and recorded using a UV–visible spectrophotometer (Thermo Scientific, Waltham, MA, USA, Genesys 10S). The β-carotene standard was used for quantification, and the results were expressed as µg β-carotene/g sample [27]. The test was conducted in triplicate.

The retention after encapsulation was calculated as a ratio of the total carotenoid amount in the microparticles to that in the feed materials before atomization.

### 2.5. Stability Study of Microparticles

The samples were placed in glass vials covered with aluminum foil and stored in desiccators containing saturated solutions of magnesium chloride MgCl_2_ (relative humidity, RH of 32.8%). The desiccators were kept at a temperature of 25 °C, and the storage period lasted for 90 days under the specified conditions [28]. The samples were evaluated every 15 days in terms of carotenoid content and color. Particle size distribution, mean diameter, moisture content, and water activity were analyzed at the initial time and after 90 days of storage, in triplicate.

#### 2.5.1. Carotenoid Degradation Kinetics

Carotenoid stability was determined by comparing the total carotenoid concentration at the initial time and over the storage of encapsulated and non-encapsulated carotenoid-rich GPE. Previous studies [29,30] have hypothesized that a first-order kinetics describes adequately the reaction of carotenoids degradation. Thereby, to investigate the stability of our samples, the degradation constant (*k*) and the half-life (t_1/2_) were determined following the first-order kinetic model, according to Equations (1) and (2).
ln*C(t)*/*C*_0_ = −*kt*(1)
t_1/2_ = ln(2)/*k*(2)
where *C* = carotenoid concentration at time *t* (μg/g); *C*_0_ = initial carotenoid concentration (μg/g); *t* = time (days).

#### 2.5.2. Color

The microparticles color defined by the parameters L* (Brightness), a* (red–green), and b*(yellow–blue) were determined using a HunterLab Mini Scan XE colorimeter (Reston, VA, USA). *Chroma* (color saturation) was calculated according to the Minolta procedure [31]:(3)$$\text{Chroma}=\sqrt{{\left(a^{*}\right)}^{2}+{\left(b^{*}\right)}^{2}}$$

The total color difference (Δ*E*) was calculated according to Equation (4).
(4)$$\Delta E=\sqrt{{\left(L_{f}^{*}-L_{0}^{*}\right)}^{2}+{\left(a_{f}^{*}-{\;a}_{0}^{*}\right)}^{2}+{\left(b_{f}^{*}-{\;b}_{0}^{*}\right)}^{2}}$$
where ΔE  = total color difference; $L_{f}^{*}\;$ = final L*; $L_{0}^{*}\;$ = initial L*; $a_{f}^{*}$ = final a*; $a_{0}^{*}$ = initial a*; $b_{f}^{*}$ = final b*; $b_{0}^{*}$ = initial b*. 

#### 2.5.3. Mean Diameter

Mean diameter of particles were measured using the SALD-201V laser diffraction particle analyzer, Shimadzu (Kyoto, Japan). Ethanol was used as a dispersing liquid. The measurements were conducted at 25 °C.

#### 2.5.4. Moisture Content and Water Activity

The moisture content of the microparticles was determined in a moisture analyzer model MB 35 from Ohaus (Parsippany, OH, USA), in triplicate. The results are expressed in percentage. The determination of water activity (a_w_) was achieved by direct reading, on an Aqualab hygrometer, model CX-2T, from Decagon Devices Inc., Pullman, WA, USA. The readings were performed at 25 °C.

### 2.6. Dynamic Vapor Sorption (DVS) of Microparticles and Oat Flakes

Water vapor sorption isotherms of the microparticles and oat flakes were determined at 25 °C using a Dynamic Vapor Sorption instrument (Surface Measurement Systems Ltd., Allentown, PA, USA). Under a continuous airflow (200 mL/min), the system was pre-equilibrated at 5% relative humidity (RH). The samples were exposed sequentially to different relative humidities (RH) from 30 to 95%. The RH transitions and mass variations of the sample were monitored continuously. The moisture sorption isotherms were determined using the DVS Analysis Macro V6.1 software, in duplicate.

### 2.7. Thermal Properties of Oat Flakes and Oat Flakes Containing Microparticles

The thermal behavior of oat flakes and oat flakes containing microparticles was described by the parameters ‘onset of gelatinization’ (To), ‘peak gelatinization temperature’ (Tp), and ‘gelatinization enthalpy’ (J/g) and measured by a Multi-Cell Differential Scanning Calorimeter (MC-DSC, TA Instruments, New Castle, DE, USA) equipped with the TRIOS software (TA Instruments, New Castle, DE, USA). The samples dispersed in water (with a moisture content of 80%) were weighed into ampoules and sealed. An empty pan was used as the reference. The samples were equilibrated at 5 °C and then heated to 140 °C at a rate of 1 °C/min. Each sample was run in triplicate, and the average results are shown.

### 2.8. Pasting Properties

The oatmeal paste was prepared using the Discovery Hybrid Rheometer 3 (DHR-3, TA Instruments Ltd., New Castle, DE, USA) and a pasting cell geometry to simulate operating conditions where temperature and shear are applied during the process. The experiment was conducted by mixing (i) oat flakes and distilled water (80% *w*/*w*); (ii) oat flakes blended with microparticles (6% *w*/*w*), and distilled water (80% *w*/*w*); (iii) oat flakes, GPE (6% *w*/*w*), and distilled water (80% *w*/*w*).

The samples were prepared using the following procedure: (1) conditioning the sample at 25 °C for 2 min; (2) heating ramp to selected temperatures of 70, 80 or 90 °C at 5 °C/min with a shear rate of 50 or 100 1/s; (3) flow peak hold at the selected temperature and shear conditions for 120 s; (4) cooling ramp to 25 °C at 5 °C/min; (5) oscillation frequency at 25 °C from 0.01 to 10 Hz with a 0.5% strain. The testing parameters and treatments are shown in Table 1. Each sample was run in triplicate.

### 2.9. Retention of β-Carotene, Lutein, and Total Carotenoid Content in the Oatmeal Paste

Total carotenoid content of oatmeal paste containing encapsulated and free GPE was analyzed using spectrophotometry, as described in Section 2.4. The main carotenoids extracted from guaraná peels were β-carotene and lutein [24]. The contents of the incorporated carotenoids were quantified in the oatmeal paste and the raw material by HPLC to evaluate the effect of the process on the retention of these compounds, in duplicate. Carotenoid extraction was carried out following Kopec et al. [32], with some modifications according to the solvents used. Sequential extractions were performed using methanol and a mixture of ethanol: acetone: hexane (1:1:1, *v*/*v*/*v*). The extract was then injected into the HPLC instrument Agilent 1260 ultra-high-performance liquid chromatograph with a diode array detector (UHPLC-DAD), using a C30 column (YMC Inc., Meridian, ID, USA, 4.6 × 250 mm, 3 μm) [33]. The contents of β-carotene and lutein were calculated from their peak areas in comparison to standards with known concentration, using a calibration curve. Carotenoid retention was determined by comparing their content before and after the pasting process and expressed as a percentage.

### 2.10. Statistics

The data were analyzed using ANOVA and Tukey’s test in SAS statistical software (version 8.02, Statistic Analysis System). Significant differences were defined at *p* < 0.05. All data were expressed as the means ± standard deviation (SD). The pasting process was performed in duplicate.

## 3. Results and Discussion

### 3.1. Total Carotenoid Content of Microparticles

Microparticles rich in carotenoids were produced by spray drying using three formulations to assess the best treatment for further applications. Total carotenoid content and retention after encapsulation are presented in Table 2. The SD1:2 treatment showed a significantly higher carotenoid content compared to the others. This difference was consistent since the microparticles containing more carotenoid-rich extract in the proportion core:carrier material would present higher content of carotenoid.

Regarding the protective effect of carrier material on carotenoid stability after encapsulation, the SD1:4 treatment had the lowest retention. Although this sample had the highest carrier material proportion, the atomization temperature was not sufficiently high to evaporate the water and dry the microparticles effectively. Consequently, this caused the adherence of semi-moist powder to the wall of the drying chamber, which may lead to the additional exposition to the temperatures of the process, leading to degradative reactions of the carotenoids.

Carotenoids are sensitive compounds and well documented, and the temperature applied for the atomization can influence their stability. The high retention observed in SD1:2 and SD1:3 was probably associated with the relationship between the suitable degree of heat treatment and feed material of these formulations during the process. This result formalizes our hypothesis that among the main factors that affect carotenoid retention during encapsulation by spray drying is the core:carrier material proportion associated with the operational condition of atomization.

### 3.2. Stability Study of Microparticles during Storage

#### 3.2.1. Carotenoid Retention

Figure 1 shows the retention of total carotenoids in the microparticles and free GPE during storage under dark at 25 °C. Following 90 days, a carotenoid loss of 45% was found in the free GPE, whereas only ~30% was observed in the microparticles. It is essential to highlight that free GPE, considered a control, contains sunflower oil, as mentioned in Section 2.2. Sunflower oil was added to alleviate significant losses observed in obtaining the extract, which certainly protected the carotenoids during storage.

Regarding the microparticles, the carrier material acts as a physical barrier against environmental conditions, and the core was efficiently entrapped by the gum arabic. This reduced the pronounced degradation of the active component observed in free GPE. However, isomerization and/or oxidation of these compounds were observed in the encapsulated samples. In this system, the high degree of carotenoid unsaturation concomitant with the presence of oxygen was crucial to their degradation over time.

First-order kinetics was followed to assess the stability of carotenoids in encapsulated and non-encapsulated GPE, and the results are reported in Table 3. In a first-order mechanism, the reaction rate depends on variations in the amount of only one reactant; in this case, the reactant was the carotenoids. All microparticles showed low values for the first-order rate constant (*k*), indicating a reduced degradation, as evidenced by the half-life (t_1/2_). It can be noted that the loss of carotenoids occurred faster in the non-encapsulated extract than those in the encapsulated. The findings denote that the spray drying technique improved the stability of GPE due to the protective potential of the carrier material under storage conditions, promoting a longer shelf life. Similar results were reported when carotenoid stability was evaluated in spray-dried samples [20,34].

#### 3.2.2. Color Parameters

The powder’s color, measured by the parameters L*, a*, b*, *Chroma*, and *ΔE* (total color difference), may indirectly indicate the degradation of carotenoids during storage. Overall, the color characteristics of the samples were influenced by core:carrier material concentration in the feed material formulations (Figure 2). Comparing the treatments, an increase in the L* values was observed by increasing the ratio of gum arabic solution in the samples following the order: SD14 > SD1:3 > SD1:2. Further, the formulations with the highest concentration of carotenoid-rich extract showed higher values of a* and b*, as expected. Chroma represents color saturation, which was in the first quadrant of the CIELab chart, corresponding to the vivid red-yellow color. A slight reduction in Chroma was observed in all formulations during storage, indicating loss of color intensity and degradation of the pigment, which corroborates with the results found in the stability study.

Regarding the trend for the total color difference, the expression of *ΔE* values for SD1:2 (7.31), SD1:3 (6.19), and SD1:4 (5.19) exhibited the overall variation between samples at initial time and samples stored 90 days in the absence of light. This result is related to the degradation of carotenoids during the period, considering stability to the component of interest conferred by the carrier agent (gum arabic). Indeed, the color variation over storage had some influence on the powder quality, considering the color property as well as the bioactivity capacity of these pigments.

#### 3.2.3. Mean Particle Diameter

Particle size is an important property for application as an ingredient in food products. Particle size distribution can be influenced by external and/or internal factors. The conditions used in the spray drying procedure, such as temperature, pressure nozzle, airspeed, and feed flow, are some of the external factors. However, the feed material formulation and its preparation process may act as the internal factors [35].

Unlike formulations SD1:2 and SD1:3, SD1:4 microparticles exhibited a remarkable size variation, in which the median diameter ranged from 9 to 18 μm, over storage (Table 4). This phenomenon was attributed to agglomeration. The hydrophilic active sites of gum arabic (carrier material) can absorb water-favoring adherence properties of the samples. Due to the electrostatic effects and covalent bonds, adhesion between the wetted microparticles occurs, and the contact or collision among them can generate new agglomerate structures [36,37]. Considering the higher proportion of gum arabic in the SD1:4 formulation, the effect of this phenomenon was maximized and reflected in the size variation after 90 days.

The particle size was within the typical range for atomization, which varies from 5 to 150 µm. Their dimensions were below the value (<100 μm) of particles which have been found to cause little or no interference sensory when being added to food. Moreover, this parameter may be associated with bioavailability and solubility of the active components entrapped, considering that the more surface area of the particle the more the bioactivity of the compounds [38].

#### 3.2.4. Moisture Content and a_w_

Moisture and a_w_ are indicators of drying efficiency and are the main factors that affect powder stability. The mean values for these parameters before and after storage are shown in Table 5.

After 90 days, the moisture and water activity of the powders increased significantly (*p* < 0.05). This was due to the water uptake of the powder during storage under a relative humidity of 33% and 25 °C. However, even after this period, the samples showed low values of water activity. The moisture content and water activity can influence microbial growth, in addition to enabling biochemical reactions. According to the literature, to avoid microbial growth, the water activity of products must be less than 0.6 [18,39]; therefore, the powders may be considered microbiologically safe. Further, a_w_ values near 0.3 imply that the food products are less sensitive against non-enzymatic browning and enzymatic activities during storage [40,41].

The relevance of powdery ingredients lies in their large application in the food industry. The incorporation of microparticles enriched with carotenoids in a model food requires investigating the characteristics of the powder in general. In this way, it would be possible to achieve a suitable application considering the convenient features for processing. The findings were expected to suggest a reasonable formulation, considering the carotenoid content and particle behavior. Thus, the formulation SD1:2 was selected to be incorporated in oatmeal paste, to better understand the effect of conditions prevalent on processing and/or preparation on carotenoid stability and paste properties.

### 3.3. Dynamic Vapor Sorption (DVS) of Oat Flakes and Microparticles

Most materials are sensitive to the presence of water vapor or moisture content in a system [42], also shown in the storage study performed in this work and described in previous section. Investigation of interactions between water and oat flakes, as well as between water and microparticles is essential to improve the conception of the effect of the encapsulated GPE incorporation on the characteristics of oatmeal pastes. As shown in Figure 3, the equilibrium moisture content of the microparticles was 5-fold higher than that of oat flakes at RH of 60%. The DVS curves for the samples were type III (non-sigmoidal).

A moisture sorption isotherm is determined by subjecting a material to different increasing relative humidity and monitoring the change in mass due to water absorption. The absorption of water depends on the number of available sites in the material capable of binding water molecules [43,44].

The structural difference between oat flakes and encapsulated GPE may be related with their distinct absorption rates. Gum arabic is a complex polysaccharide with a highly branched structure, which facilitates polar interactions with water by hydrogen bonds at room temperature [45], whereas oat flake comprises mainly starch and fiber. Starch contains amylose, a linear molecule, and amylopectin, a non-linear and highly branched molecule [46,47], both with a hydrophilic nature. In the crystalline regions of the granule, the intermolecular interactions among the chains are very strong, and the diffusion of a plasticizer, such as water, is slow at room temperature.

### 3.4. Thermal Properties

The onset temperature and the peak temperature for oat flakes and oat flakes incorporated with encapsulated GPE with a moisture content of 80% are listed in Table 6. The thermal properties were significantly affected (*p* ≤ 0.05) by the addition of microparticles, showing a decrease in the enthalpies and a slight increase in gelatinization temperatures. This indicates that in samples containing oat flakes and microparticles, the starch was less gelatinized which is likely attributed to the decreased availability of water [48,49]. In the system studied, microparticles and starch competitively bind water to form hydrogen bonds. As part of the water is bound to the microparticles with larger water absorption capacity shown in Figure 3, the available water for starch is reduced, which could elevate the starch gelatinization temperature due to insufficient hydration and swelling.

### 3.5. Pasting Properties

The viscosity of the oat flakes starch with and without GPE was investigated during the heating–cooling procedure, as described in Section 2.8, to evaluate their influence on the oatmeal paste properties. In Figure 4, gelatinization of the samples was observed at the temperature of 70 °C or above, showing a rapid increase in viscosity.

Temperature significantly affected the pasting behavior of the samples. At 70 °C and 80 °C, the viscosity of oatmeal paste with encapsulated GPE was lower without a noticeable peak. The increase in starch viscosity indicates the swelling of the starch molecules. Further heating resulted in molecular disorganization and the leaching of amylose and amylopectin into the solvent. Shear facilitated the formation of the paste by promoting non-covalent interactions between starch molecules, which can be affected by temperature and sample composition [50,51].

Microparticles produced using gum arabic as the carrier material hindered the swelling/gelatinization of the grains. Singh, Geveke, and Yadav [52] suggested that gum arabic might cover the surface of the starch during processing, leading to a reduced interaction among neighboring starch granules, which may control the swelling power of starch and restrict the increment of viscosity. In this case, the system would require higher temperatures to completely gelatinize the starch. Shahzad et al. [53] reported similar findings.

At 80 °C, the oatmeal paste with free GPE showed significantly higher peak viscosity than the control (only oat flakes and water). This may be attributed to the presence of residual ethanol from GPE in the starch granules, which favors maximum swelling and leads to the gelatinization of most of the starch in the system [54]. Previous researchers [55] have reported that an aqueous ethanol medium promoted the formation of hydrogen bonds with starch molecules, resulting in stronger gels than those prepared with pure water. Alternatively, GPE contains a high amount of carotenoids. These compounds are hydrophobic, which may interact with double helices structures via hydrophobic interactions, and contributes to higher viscosity.

Singh et al. [54] suggested that gum arabic might cover the surface of the starch during processing, leading to the reduced interaction among neighboring starch granules, which may control the swelling power of starch and restricted the increment of viscosity.

At 80 °C, the oatmeal paste with free GPE showed significantly higher peak viscosity than the control. As before, this is attributed to the presence of residual ethanol in the extracted GPE, which favors maximum swelling and leads to the gelatinization of most of the starch in the system [56]. Previous researchers [57] have reported that an aqueous ethanol medium promoted the formation of hydrogen bonds with starch molecules, resulting in stronger gels compared to those prepared with pure water.

### 3.6. Carotenoid Retention

The formulation of foods enriched with active compounds is a matter of concern for the food industry, considering the impact of processing on its bioactivity. For this reason, the retention of β-carotene and lutein under the thermomechanical treatment applied during the production of oatmeal paste was investigated. Results are illustrated in Figure 5.

The behavior of β-carotene revealed higher stability under temperature and shear conditions applied during the paste preparation, regardless of whether it was in an encapsulated or non-encapsulated form, with retention ranging from 59 to 87%. The lutein stability was significantly higher in samples enriched with encapsulated GPE compared to the sample with free GPE, varying from 51 to 89%. Regarding the testing conditions, 90 °C and the shear rate of 100 1/s exhibited a larger detrimental effect on the carotenoids, either free or in their encapsulated forms.

Carotenoids are classified into carotenes and xanthophylls, and the latter are more polar. The first group, comprising pro-vitamin A carotenoids (β-carotene), demonstrated greater stability. Higher retention of β-carotene in an aqueous medium may be due to its physical stability towards environmental conditions. In contrast, the reactivity of xanthophylls, such as lutein, is highly affected during food processing due to its structure, characterized by the presence of oxygen in the molecule chain. Trapped oxygen within the food matrix composed of water and oat flakes may contribute to the oxidation of the carotenoids in the free guaraná peel extract, and as a result, this system led to the substantial loss of lutein. Therefore, the encapsulation technique can reduce the degradation of this component. Dhuique-Mayer et al. [56] assessed the stability of carotenes and xanthophylls from citrus juice upon thermal treatment, and they suggested that the former may react less in a polar solvent. Interestingly, on the opposite, when the stability of β-carotene and lutein in oil (i.e., a non-polar solvent) was investigated during heat treatment, they found that β-carotene had the highest reactivity [57].

Carotenoids have highly unsaturated chains and are easily oxidized. In addition, the isomerization of these compounds also affects their stability through food processing [58,59]. During the preparation of the oatmeal paste, extrinsic and intrinsic factors can be considered the promoters of oxidation/isomerization reactions. Thermomechanical stress, the incidence of light, and the presence of oxygen may be the main external reasons contributing to the loss of carotenoids. The composition of the food matrix, the physical status of the carotenoid, and the air incorporation into the sample may be considered the internal factors affecting the active retention [27,60].

Indeed, degrading reactions are promoted by the structure of the carotenoid as well as the factors mentioned previously. Similar results have been reported by other researchers [61,62,63].

## 4. Conclusions

Microencapsulation of carotenoid-rich extract from guaraná peels enhanced its stability over storage. The sample SD1:2 exhibited promising features among other formulations, including high carotenoid content, suitable particle size, and intense color. The addition of microparticles rich in bioactive compounds into oatmeal paste increased the onset and peak temperature of starch but decreased its enthalpy. Further, it reduced the viscosity of the system, whereas the opposite trend was observed for the samples with free GPE. Overall, the gelatinization of oat flakes starch was affected by the decrease in accessible water, due to the presence of the microparticles.

Testing conditions, such as temperature and shear, favored heat transfer and oxygen incorporation in the system, reducing the total carotenoid content of the samples. In addition, the treatment significantly altered the lutein content, when compared to β-carotene. However, pigment loss decreased by encapsulation.

The recovery of bioactive components from by-products, such as β-carotene (provitamin A) and lutein (recognized as an agent that prevents macular degeneration), reveals the importance of the present research on adding value to a waste of a food process encouraging sustainable development. In addition, the study of applications of these materials, as suggested in the current work, represents a suitable trend for obtaining functional foods. The range of oat products and derivatives is wide, including oatcake, oatmeal, and porridge, which can be enriched with bioactive compounds to add health benefits. In addition, the use of oats as an ingredient in newer sectors, such as plant proteins, has grown and has opened up possibilities for applications and, consequently, new processes.

## Figures and Tables

**Figure 1 foods-12-01170-f001:**
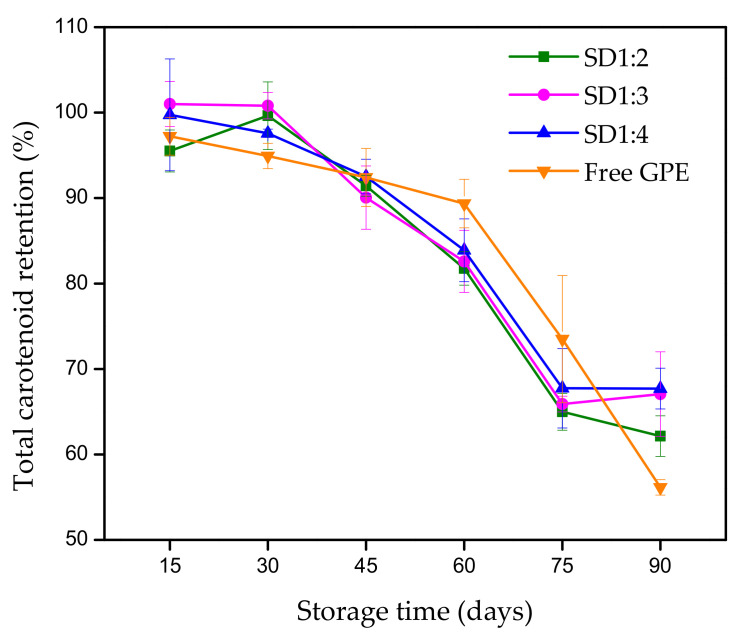
Total carotenoid retention in free and encapsulated guaraná peel extract (GPE) produced by spray drying at a temperature of 100 °C. The formulations represent the proportion of the core:carrier material.

**Figure 2 foods-12-01170-f002:**
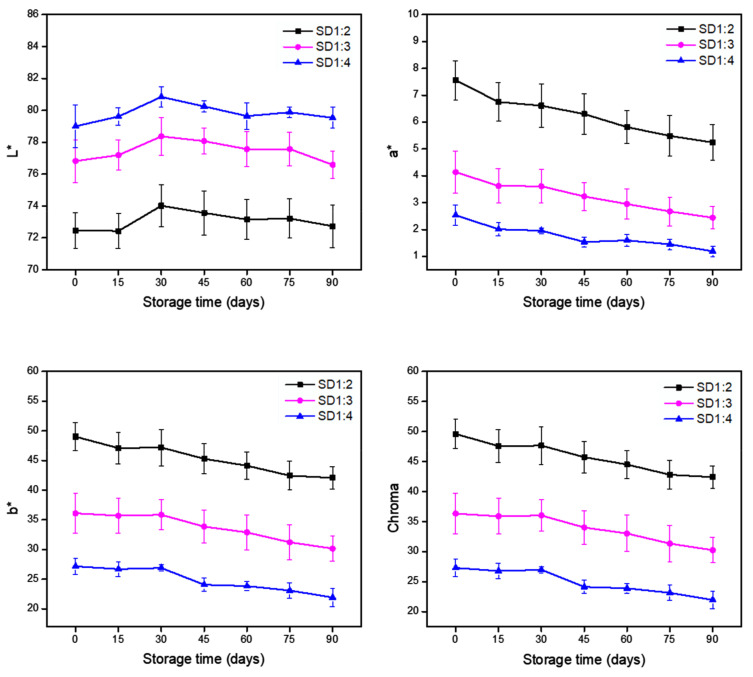
L*, a*, b*, and Chroma parameters obtained for each formulation produced by spray drying in instrumental color analysis during storage at 25 °C. The formulations denote the proportion of the core:carrier material.

**Figure 3 foods-12-01170-f003:**
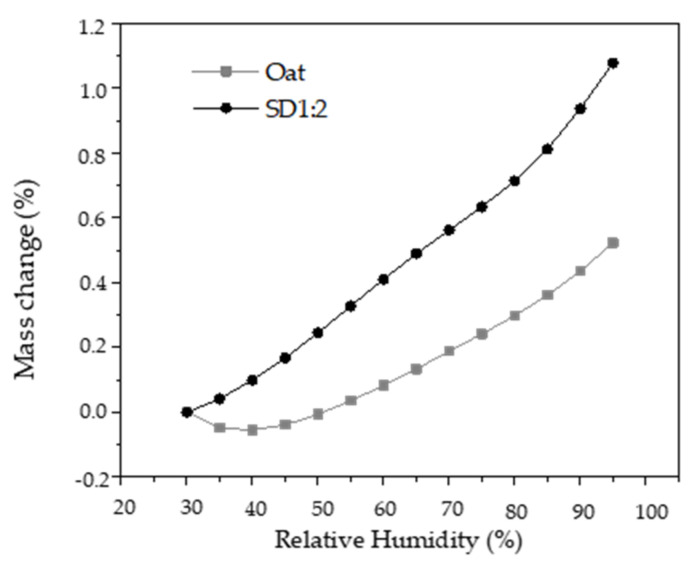
Dynamic vapor sorption isotherms of particles produced by spray drying (SD1:2) and oat flakes.

**Figure 4 foods-12-01170-f004:**
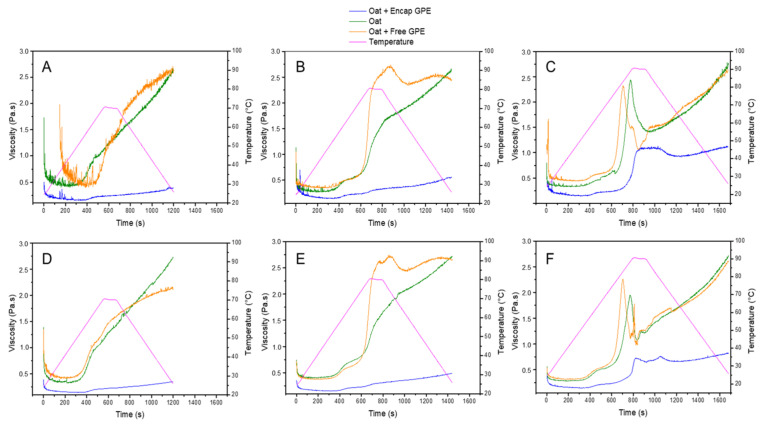
DHR-3 pasting curves of oatmeal paste enriched with encapsulated GPE and free GPE. Testing conditions: (**A**) temperature 70 °C and shear rate 50 1/s; (**B**) temperature 80 °C and shear rate 50 1/s, (**C**) temperature 90 °C and shear rate 50 1/s; (**D**) temperature 70 °C and shear rate 100 1/s; (**E**) temperature 80 °C and shear rate 100 1/s; (**F**) temperature 90 °C and shear rate 100 1/s.

**Figure 5 foods-12-01170-f005:**
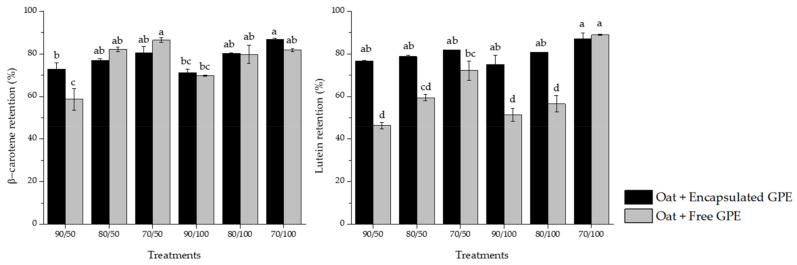
β-carotene and lutein retention in oatmeal paste incorporated with encapsulated and free guaraná peel extract (GPE) after the process at different conditions (temperature/shear rate). Different letters in the columns represent a significant difference (*p* < 0.05).

**Table 1 foods-12-01170-t001:** Conditions used for the preparation of oatmeal pastes in DHR-3.

Treatments	Parameters	
	Temperature (°C)	Shear (1/s)
70/50	70	50
80/50	80	50
90/50	90	50
70/100	70	100
80/100	80	100
90/100	90	100

**Table 2 foods-12-01170-t002:** Total carotenoid content and retention in microparticles after the spray drying (SD) process.

Formulations	Carotenoid Content (µg/g)	Carotenoid Retention (%)
SD1:2	96.0 ± 1.0 ^a^	100.0 ± 1.0 ^a^
SD1:3	57.6 ± 0.7 ^b^	99.4 ± 0.2 ^a^
SD1:4	40.7 ± 0.9 ^c^	96.7 ± 0.3 ^b^

In formulations, the name is the proportion of the core:carrier material. The results are expressed as the mean ± standard deviation (*n* = 3). Different letters in the columns represent a significant difference (*p* ≤ 0.05).

**Table 3 foods-12-01170-t003:** Kinetic parameters for carotenoid degradation in non-encapsulated and encapsulated extract during storage.

Sample	*k* (s^−1^)	t _1/2_ (Days)	R^2^
Free extract	0.006	108.082	0.811
SD1:2	0.005	141.924	0.888
SD1:3	0.005	152.286	0.897
SD1:4	0.005	144.676	0.874

Formulation SD (core:carrier material ratio).

**Table 4 foods-12-01170-t004:** Mean and standard deviation of median diameter sizes (μm) expressed in volume for each formulation of microparticles obtained by spray drying at 0 and 90 days of storage at 25 °C.

Time (days)	Formulations	Median Diameter (µm)
0	SD1:2	11.9 ± 1.5 ^a^
0	SD1:3	9.1 ± 1.8 ^ab^
0	SD1:4	8.9 ± 0.9 ^b^
90	SD1:2	15.1 ± 2.8 ^a^
90	SD1:3	13.4 ± 2.4 ^a^
90	SD1:4	18.6 ± 5.5 ^a^

Formulation SD (core:carrier material ratio). Results are expressed as the mean ± standard deviation (*n* = 3). Different letters in the columns represent a significant difference (*p ≤* 0.05).

**Table 5 foods-12-01170-t005:** Moisture and water activity (a_w_) of encapsulated carotenoid-rich guaraná peel extract.

Time (Days)	Formulations	Parameters	
		Moisture (%)	a_w_
0	SD1:2	3.9 ± 0.4 ^d^	0.21 ± 0.02 ^c^
0	SD1:3	4.8 ± 0.4 ^c^	0.24 ± 0.03 ^c^
0	SD1:4	4.1 ± 0.7 ^cd^	0.16 ± 0.02 ^d^
90	SD1:2	6.9 ± 0.3 ^b^	0.46 ± 0.01 ^b^
90	SD1:3	8.1 ± 0.3 ^a^	0.48 ± 0.01 ^a^
90	SD1:4	8.2 ± 0.1 ^a^	0.50 ± 0.02 ^a^

Formulation SD (core:carrier material ratio). The results are expressed as the mean ± standard deviation (*n* = 3). Different letters in the columns represent a significant difference (*p ≤* 0.05).

**Table 6 foods-12-01170-t006:** DSC gelatinization properties of oat flakes and oat flakes enriched with encapsulated GPE.

	Onset Temp. (°C)	Peak Temp. (°C)	Enthalpy (J/g)
Oat	53.05 ± 0.01 ^b^	60.1 ± 0.3 ^b^	3.1 ± 0.1 ^a^
Oat + Encapsulated GPE	54.3 ± 0.6 ^a^	61.0 ± 0.3 ^a^	2.6 ± 0.1 ^b^

The results are expressed as the mean ± standard deviation (*n* = 3). Different letters in the columns represent a significant difference (*p* ≤ 0.05).

## Data Availability

The data presented in this study are available upon request.

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
