# Peer review of "Microencapsulation of Carotenoid-Rich Extract from Guaraná Peels and Study of Microparticle Functionality through Incorporation into an Oatmeal Paste"

_foods, 2023, doi:10.3390/foods12061170_

Round 1
Reviewer 1 Report
In the current manuscript the carotenoid extract from guaraná peels was encapsulated by spray drying, and its effects on the pasting characteristics of oatmeal paste were evaluated.The article is well designed and has enough experiments. The results are well presented and discussed. In my opinion, the present manuscript need minor revision as follows:
Introduction:
-In the introduction, some researches that have been published in the field of encapsulation of natural carotenoids by spray dryer method should be mentioned. Use the following articles:
Etzbach, L., Meinert, M., Faber, T., Klein, C., Schieber, A., & Weber, F. (2020). Effects of carrier agents on powder properties, stability of carotenoids, and encapsulation efficiency of goldenberry (Physalis peruviana L.) powder produced by co-current spray drying. Current Research in Food Science, 3, 73-81.
Hojjati, M., Razavi, S. H., Rezaei, K., & Gilani, K. (2011). Spray drying microencapsulation of natural canthaxantin using soluble soybean polysaccharide as a carrier. Food Science and Biotechnology, 20(1), 63-69.
Tuyen, C. K., Nguyen, M. H., & Roach, P. D. (2010). Effects of spray drying conditions on the physicochemical and antioxidant properties of the Gac (Momordica cochinchinensis) fruit aril powder. Journal of Food Engineering, 98(3), 385-392.
Materials and methods:
-In what season were the fruits prepared?
-Write the details of the sunflower oil production company.
-Why did you use 3% concentration of oil?
L108: Is aluminum foil correct?
L126-7: Define the color indices a and b correctly.
Results and Discussion:
-All numbers should be written with one decimal place (Table 2 and 4).
-aw is correct. Please check and rewrite in the text.
-All numbers of aw should be written with two decimal places (Table 5).
-Indicate statistical differences between treatments by placing different lowercase English letters on the columns (Figure 5).
Conclusion:
-What uses do you suggest for the product produced in this research?
Author Response
We appreciate the reviewer’s time and suggestions on the manuscript. The manuscript has been revised and highlighted in red in the text. The point-to-point can be seen attached.

Reviewer 2 Report
In the introduction, the authors could have provided more justification for the research objective undertaken. At this point are descriptions of carotenoids, agro-waystes and generally used methods for starches of various origins.
And why exactly oatmeal paste? Why guarana extract? And why it together?
The methodology is described correctly with basic experimental data and allows its subsequent use by scientists.
The results and their discussion are carried out carefully and correctly.
Correct conclusions.
Author Response

(The authors gave the same response as above.)

Reviewer 3 Report
In this study authors used spry drying to encapsulate the carotenoid-rich extract from guaraná peels and characterize the microparticles and their influence on the pasting properties and carotenoids stability of oatmeal paste. I have some remarks and questions concerning this manuscript.
Line 109: Please give “2” in MgCl2 in subscript.
Line 119 the reaction r. What does r mean in this sentence “r”?
Please explain all variables used in equation 5.
Line 126-127 „say what color represents”???
Please explain the meaning of the variables in equation 5.
For a few tests, the number of repetitions was included whereas for the others not. Please complete this information. Why standard deviation is written in capital letters?
Fig. 2. For b* and Chrome the same charts with the same values were included.
Please enlarge the size of Fig. 5. And include the significant differences between means.
Delete equation 1. The description in lines 103-106 is sufficient.
Point 3.2.4. Please define Aw before the first time use.
Fig. 5. Please include the higher size of figures with marked significant differences between means.
Author Response

(The authors gave the same response as above.)

Round 2
Reviewer 3 Report
The authors corrected the menauscript accordingly.